# Dosing patterns and dose effects of sacubitril/valsartan: A claims-based retrospective cohort study

Jillian M. Rung[1]*, Tyson S. Barrett[1], Keith LeJeune[1,2], Shannon B. Richards[1], Amresh Raina[3], Lawrence Sinoway[4,5]

**1** Enterprise Analytics, Highmark Health, Pittsburgh, Pennsylvania, United States of America, **2** Allegheny Health Network, Pittsburgh, Pennsylvania, United States of America, **3** Allegheny General Hospital, Pittsburgh, Pennsylvania, United States of America, **4** Penn State Health, Hershey, Pennsylvania, United States of America, **5** Penn State University, University Park, Pennsylvania, United States of America

* jillian.rung@highmarkhealth.org

## Abstract

### Purpose

The goals of this retrospective cohort analysis were to determine real-world dose and titration patterns of sacubitril/valsartan (SAC/VAL), a heart failure medication, and examine whether dose patterns are associated with healthcare utilization and costs.

### Methods

Adult health plan members (18-100 years old) who initiated SAC/VAL between 2020 and 2022 and had continuous enrollment 6 months prior to, and at least 3 months following SAC/VAL initiation were identified. Members also had to have 3 months of SAC/VAL fills with good adherence ($N = 2,977$). Claims data were used to characterize dosage patterns and compare total costs of care, as well as all-cause and heart failure- hospital admissions across those with different terminal SAC/VAL doses.

### Results

Most members initiated SAC/VAL at the lowest dose (76%, $n = 2,267$), of whom few titrated upward by their final fill (31%, $n = 703$). Overall, only 19% ($n = 563$) were at target by their final fill. Those ending on higher doses experienced significantly fewer all-cause admissions (incidence rate ratios of 1.52 [$SE = .16$] to 2.66 [$SE = .37$]; $p$s < .001) and incurred significantly lower total costs of care while on SAC/VAL (cost ratios of 1.21 [$SE = .06$] to 1.48 [$SE = .09$]; $p$s < .001).

### Conclusion

Most individuals initiate and remain on the lowest SAC/VAL dose despite guidelines to titrate upward. SAC/VAL dosage is significantly associated with outcomes, with higher doses associated with more clinical and cost benefits. Research is needed to identify barriers to dose titration and to develop interventions for maximal patient benefit.

**Data availability statement:** The data cannot be made publicly available due to the presence of sensitive and potentially identifiable information; doing so would risk confidentiality and privacy of members in the study sample. Therefore no data beyond the aggregate are available to be shared. Additional information on limitations of data sharing can be obtained from the Allegheny Health Network Research Institute via AHNIRB@ahn.org or 1-844-577-4621.

**Funding:** The author(s) received no specific funding for this work.

**Competing interests:** The authors have declared that no competing interests exist.

# Introduction

Sacubitril/valsartan (SAC/VAL, Entresto) is a highly effective combination angiotensin receptor-neprilysin inhibitor (ARNi) that is used to treat heart failure (HF). Foundational work from the PARADIGM-HF trial demonstrated that 200 mg twice daily (bid) of SAC/VAL reduced hospitalizations and cardiovascular- and all-cause mortality [1] compared to enalapril (10 mg bid), an angiotensin-converting enzyme inhibitor (ACEi) that is also effective for reducing mortality in HF patients [2]. Thus, the 2022 HF guidelines recommend using ARNi before an ACEi or angiotensin receptor blockers (ARBs) [3].

While SAC/VAL's effectiveness and superiority was based on 200 mg bid data, subsequent real-world data suggest that most patients take a lower dose. Data from one HF registry show that only 14% of patients on SAC/VAL were at the target dose, and 57% were at a dose of < 50% target [4]. Moreover, a post-hoc analysis of PARADIGM-HF revealed that those whose dose was reduced from the target had a higher incidence of composite cardiovascular deaths and first HF hospitalization [5]. Kido et al. more recently demonstrated that hospitalization rates were higher for the lowest (50 mg) than for the two higher SAC/VAL doses (100 and 200 mg, respectively) [6]. Similarly, Chen et al. demonstrated that "low dose" SAC/VAL was associated with greater HF hospitalizations and all-cause mortality than "high dose" [7].

To better understand discrepancies between dosing guidelines and real-world practice, we used data from a large health plan to examine a series of issues pertaining to SAC/VAL dosing. With a sample of nearly 3,000 members initiating SAC/VAL, we identified the most commonly prescribed dose and dose trajectories and explored causes of filling the lower than maximal dose. Finally, we quantified the hospitalization and cost impact of guideline-discordant use of SAC/VAL.

# Materials and methods

## Data source and health system

Our primary data source was an administrative database containing medical, pharmacy, and vision insurance claims for enrollees of Highmark Inc. health plans. Only approved claims were used in analyses. Between January 1, 2019 and December 31, 2022, the claims database contained between 5 to 7 million members, the majority of whom lived in Pennsylvania, Delaware, West Virginia, and New York, and a minority within the remaining 46 states. Highmark Inc. is a part of a blended-health organization owned by Highmark Health, which includes a hospital and provider network (Allegheny Health Network; AHN). Electronic medical records were accessed for a small, randomly selected subset of health plan members included in the final sample who also received care within AHN for the purposes of data validation (details provided below). Data from medical records were not included in analyses reported herein. Study data also included publicly available location-based metrics from the CDC (e.g., Social Vulnerability Index [SVI]) [8]. Data were accessed from the database on February 1st, 2024, with additional information for secondary analyses (reported in discussion) accessed on October 3rd, 2024. Authors had access to information that could identify members during and after data collection. Additional aggregated, de-identified data that support this study's finding can be provided upon reasonable request by contacting the corresponding author.

## Sample inclusion and exclusion criteria

Eligible members between 18 and 100 years of age had approved claims for SAC/VAL between January 1st, 2019 and December 31st, 2022 with the earliest fill occurring between January 1st, 2020 and December 31st, 2022; the earliest fill served as a members' index date. Fills from 2019

were included to ensure the index date was either the member's first exposure to SAC/VAL or their first after a significant wash-out. Members also needed continuous medical and prescription coverage 6 months prior to their first fill of SAC/VAL and for at least 3 months following, as well as 90 days' supply or more of SAC/VAL filled.

Fill dates and days supplied of SAC/VAL were adjusted to account for overlap between the fills, which is consistent with methodology for calculating the proportion of days covered (PDC) [9]. These adjusted dates were used to determine the time span that members took SAC/VAL. The end of members' follow-up was assigned based on what came first: death (when this information was available), the end of continuous enrollment, or the end of their time taking SAC/VAL. Using these data, the PDC was also calculated for each member. PDC ranges from 0 (no days covered) to 1 (all days covered). Only members who continuously filled SAC/VAL during follow-up (PDC of 0.75 to 1.0) were included because compliance may affect dosing regimens and outcomes. Other exclusionary conditions were missing demographic data, metastatic cancers, moderate to severe liver disease, dementia, AIDs, undergoing dialysis, or pregnancy during baseline or follow-up. Fig S1 in S1 File shows the number of members excluded by these criteria.

### Demographics and medical characteristics

Demographic information (e.g., age, sex) and members' residence (location) was obtained from the claims database. Location was used to append SVI [8]. Health status was characterized over the 6 months preceding the index date using the Charlson Comorbidity Index (CCI)[10,11] and the presence/history of additional, relevant conditions and procedures (e.g., hypertension, percutaneous coronary intervention). Members' duration of HF and evidence of reduced ejection fraction were also obtained. Heart failure duration was based on the duration from the earliest HF diagnosis claim to the index date. Evidence of reduced ejection fraction was defined as having at least one non-diagnostic claim with an ICD-10 CM code associated with mild to reduced ejection fraction (I50.82, I50.84, I50.2x, or I50.4x; see Sandhu et al. [12]) on or before the index date (see Supplemental Methods and Table S1 in S1 File for details). Ejection fraction was inferred from the presence of relevant diagnosis codes on claims due to the lack of available test results for the sample (such data are not typically included within claims since they are not necessary for reimbursement).

### Prescription fills

All prescription claims for SAC/VAL, ACEi, ARBs, beta blockers, and sodium glucose cotransporter-2 (SGLT2) inhibitors during the baseline and follow-up periods were identified. A list of the drug names included for each class and the National Drug Codes (NDC) for SAC/VAL fills are provided in Tables S2 and S3 in S1 File. The fields extracted from pharmacy claims were fill date, NDC, dosage (low, 24/36 mg; medium, 49/51 mg; high, 97/103 mg), and days supplied. This information was summarized in various ways (e.g., percentage of members with lowest dose for first fill, number of dose changes per-member) to develop descriptive categories of dosage trajectories.

### Costs and utilization

The total cost of care was calculated using the negotiated costs per service ("allowed amounts") across all claims during baseline and follow-up for each member. All-cause and HF hospital admissions were identified using claim type (e.g., inpatient vs. professional claims), place of service, and diagnosis codes from individual claims. See Supplemental Methods in S1 File for coding details.

## Analyses

Dosing-related prescribing patterns were summarized descriptively and graphically, from which the most differentiating dose-related grouping definition was chosen for subsequent analyses (i.e., dosage of final fill). Selected prescription fill metrics and categorizations are presented to characterize the predominant dosage patterns across fills. To identify characteristics that differed across dose groups, Kruskal-Wallis, Fisher's Exact, and chi-square tests were conducted as appropriate, and corresponding effect sizes were calculated (eta-squared and Cramer's *V*). Differences in total costs of care and hospital admissions (all-cause and HF-related) as a function of SAC/VAL dose were analyzed using generalized linear models with gamma (costs), or negative binomial or Poisson (admissions) error distributions. For all models, demographic and medical characteristics that differed across dose groups ($p \leq .10$) were included as covariates, as were outcome-specific baseline measures (e.g., baseline all-cause admissions for the model predicting all-cause admissions during follow-up). High-influence observations were excluded from models of costs and all-cause admissions; a summary of results from these models with all observations is included in Supplemental Results and Table S4 in S1 File. All tests were two-sided and used an alpha of.05. For costs and counts of admissions, we performed sensitivity analyses by replicating the models only among those with the shortest and longest durations on SAC/VAL (3-8 and 18 + months) to evaluate initial versus sustained differences. Additional details pertaining to modeling, diagnostics, covariates, and model estimates are included in Supplemental Methods in S1 File. All data processing and analyses were conducted using *R* version 3.6.3.

For the purposes of SAC/VAL dosing data validation, a random sample of 12 members who had claims for encounters within Allegheny Health Network was identified. Selection was performed such that half of the members' initial and terminal SAC/VAL fills were at the lowest dose (i.e., these members only ever filled the lowest dose), and the other half evidenced an increase in dose from their initial to terminal fill. Manual review of each selected member's medical record was performed to verify if the dosing information identified via claims matched. There was 100% concordance between the dosing information in claims and members' medical records.

The Allegheny Health Network IRB reviewed this research (protocol 2023-142) and found it qualifies for exempt status according to the following category in the Code of Regulations: 45 CFR 6.104 (d) Category (Exempt 4 - Secondary Research without Consent - Research involves the secondary use of identifiable private information or identifiable specimens) with a "Waiver of Informed Consent". This research was conducted in accordance with the principles of the Declaration of Helsinki. Reporting follows EQUATOR standards (STROBE), as outlined in S1 File (page numbers are based on original submitted manuscript).

## Results

The sample included 2,977 members with a median age of 67 years (Table 1). The majority were male (69%), and most members were covered by Medicare Advantage (47%) or commercial plans (44%). A majority were from the northeast region of the US (74%), but all regions were represented. Most members had comorbid conditions, with a median CCI score of 3.0 (e.g., three low-severity conditions or one severe condition). The middle 50% of members had a score of 2.0 to 4.0, which reflects mild to moderate comorbid diseases. The most common CCI condition prior to SAC/VAL initiation was congestive HF (89%), followed by peripheral vascular disease (36%). Between 22% and 24% of members had diabetes with complications, renal disease, or chronic pulmonary disease. Among the additional conditions and procedures, dyslipidemia (73%) and hypertension (78%) were common, and 39% had atrial fibrillation (see Table S5 in S1 File for additional details).

**Table 1. Demographic, enrollment, and select medical characteristics for the full sample and by dose groups.**

| Characteristic | Overall N = 2,977[a] | Final Dose Filled | | | p-value[b] | Effect size |
|---|---|---|---|---|---|---|
| | | 24/26 mg n = 1,605[a] | 49/51 mg n = 809[a] | 97/103 mg n = 563[a] | | |
| Age | 67 (58, 77) | 69 (60, 79) | 65 (57, 76) | 63 (56, 73) | < .001 | 0.026 |
| Sex | | | | | .001 | 0.068 |
| Female | 930 (31%) | 543 (34%) | 242 (30%) | 145 (26%) | | |
| Male | 2,047 (69%) | 1,062 (66%) | 567 (70%) | 418 (74%) | | |
| Index Year | | | | | < .001 | 0.091 |
| 2020 | 705 (24%) | 337 (21%) | 189 (23%) | 179 (32%) | | |
| 2021 | 977 (33%) | 490 (31%) | 296 (37%) | 191 (34%) | | |
| 2022 | 1,295 (44%) | 778 (48%) | 324 (40%) | 193 (34%) | | |
| Insurance Type | | | | | < .001 | 0.109 |
| Medicare Advantage | 1,399 (47%) | 858 (53%) | 343 (42%) | 198 (35%) | | |
| Commercial | 1,320 (44%) | 620 (39%) | 392 (48%) | 308 (55%) | | |
| Individual | 137 (4.6%) | 63 (3.9%) | 41 (5.1%) | 33 (5.9%) | | |
| Medicaid | 82 (2.8%) | 39 (2.4%) | 24 (3.0%) | 19 (3.4%) | | |
| Other | 39 (1.3%) | 25 (1.6%) | 9 (1.1%) | 5 (0.9%) | | |
| Months of follow-up | 12 (6, 21) | 10 (6, 18) | 12 (7, 21) | 16 (9, 26) | < .001 | 0.039 |
| Region | | | | | .074 | 0.044 |
| Midwest | 147 (4.9%) | 70 (4.4%) | 44 (5.4%) | 33 (5.9%) | | |
| Northeast | 2,216 (74%) | 1,233 (77%) | 587 (73%) | 396 (70%) | | |
| South | 585 (20%) | 288 (18%) | 170 (21%) | 127 (23%) | | |
| West | 29 (1.0%) | 14 (0.9%) | 8 (1.0%) | 7 (1.2%) | | |
| Evidence of Reduced EF | 1,929 (65%) | 1,014 (63%) | 526 (65%) | 389 (69%) | .040 | 0.046 |
| Duration of HF (days) | 71 (20, 154) | 76 (20, 156) | 73 (21, 152) | 56 (23, 146) | .459 | 0.000 |
| Unknown | 247 | 136 | 65 | 46 | | |
| CCI | 3.00 (2.00, 4.00) | 3.00 (2.00, 4.00) | 3.00 (2.00, 4.00) | 3.00 (2.00, 4.00) | .244 | 0.000 |
| Index During Admission | 406 (14%) | 246 (15%) | 88 (11%) | 72 (13%) | .009 | 0.056 |
| Median Allowed PMPM (Baseline) | 655 (194, 1,733) | 721 (227, 1,772) | 565 (169, 1,610) | 541 (149, 1,798) | .002 | 0.003 |
| Any Baseline HF Admission | 576 (19%) | 304 (19%) | 150 (19%) | 122 (22%) | .293 | 0.029 |
| Any Baseline All-Cause Admission | 1,372 (46%) | 761 (47%) | 355 (44%) | 256 (45%) | .246 | 0.031 |

[a]Median (IQR); n (%)

[b]Kruskal-Wallis rank sum test; Pearson's Chi-squared test.

CCI = Charlson Comorbidity Index. EF = ejection fraction. HF = heart failure. PMPM = per-member per-month

Note. Duration of heart failure could not be calculated for some individuals due to the lack of claims containing relevant diagnosis codes within the timeframe searched.

## Dose trajectories

Most members initiated SAC/VAL at the lowest dose (76%), and a small percentage initiated at the highest dose (3.8%). More than half of members (64%) remained on the same dose for their entire duration on SAC/VAL, with any dose titration being less common. Detailed descriptive statistics pertaining to fill characteristics and dosages are shown overall and as a function of one's final dose in Table 2.

Members' dose trajectories were additionally categorized according either to evidence of both up and downward titration across fills ("Both"), or the dosage of their starting relative to final fill (e.g., increase from first to final), which is shown in Fig 1. Panel A of Fig 1 highlights that most members remained on their starting dose. Of relevance to guideline directions, only 31% and 36% of those starting on the lowest and medium doses titrated

**Table 2. Descriptive statistics and comparisons of metrics pertaining to SAC/VAL fills overall and by dose groups.**

| Characteristic | Overall N = 2,977[a] | Dose of Final Fill | | | | Effect Size |
|---|---|---|---|---|---|---|
| | | 24/26 mg n = 1,605[a] | 49/51 mg n = 809[a] | 97/103 mg n = 563[a] | p-value[b] | |
| Number of Fills | 8 (4, 13) | 6 (4, 12) | 8 (4, 13) | 10 (6, 18) | <.001 | 0.042 |
| Days Covered | 315 (173, 570) | 270 (150, 508) | 337 (180, 592) | 439 (240, 728) | <.001 | 0.041 |
| Proportion of Days Covered | 0.99 (0.93, 1.00) | 0.98 (0.92, 1.00) | 0.99 (0.93, 1.00) | 0.99 (0.94, 1.00) | .034 | 0.002 |
| Number of Dose Switches | 0.00 (0.00, 1.00) | 0.00 (0.00, 0.00) | 1.00 (0.00, 1.00) | 1.00 (1.00, 2.00) | <.001 | 0.421 |
| Any Dose Increase | 1,031 (35%) | 80 (5.0%) | 487 (60%) | 464 (82%) | <.001 | 0.692 |
| Any Dose Decrease | 253 (8.5%) | 114 (7.1%) | 82 (10%) | 57 (10%) | .013 | 0.054 |
| All Doses Same | 1,908 (64%) | 1,491 (93%) | 318 (39%) | 99 (18%) | <.001 | 0.667 |
| Dose of First Fill | | | | | <.001 | 0.438 |
| 24-26 mg | 2,267 (76%) | 1,564 (97%) | 454 (56%) | 249 (44%) | | |
| 49-51 mg | 597 (20%) | 34 (2.1%) | 350 (43%) | 213 (38%) | | |
| 97-103 mg | 113 (3.8%) | 7 (0.4%) | 5 (0.6%) | 101 (18%) | | |
| Modal Dose | | | | | <.001 | 0.807 |
| 24-26 mg | 1,786 (60%) | 1,566 (98%) | 174 (22%) | 46 (8.2%) | | |
| 49-51 mg | 740 (25%) | 29 (1.8%) | 629 (78%) | 82 (15%) | | |
| 97-103 mg | 451 (15%) | 10 (0.6%) | 6 (0.7%) | 435 (77%) | | |
| Percent of Fills at Final Dose | 1.00 (0.78, 1.00) | 1.00 (1.00, 1.00) | 0.86 (0.57, 1.00) | 0.73 (0.50, 0.91) | <.001 | 0.402 |
| Baseline ACEi Use | 1,176 (40%) | 606 (38%) | 332 (41%) | 238 (42%) | .098 | 0.040 |
| Baseline ARB Use | 949 (32%) | 459 (29%) | 290 (36%) | 200 (36%) | <.001 | 0.076 |

[a]Median (IQR); n (%)

[b]Kruskal-Wallis rank sum test; Pearson's Chi-squared test. Effect sizes are eta-squared (for Kruskal-Wallis tests) and Cramer's V (for Chi-squared tests)

upward, respectively. Members who started on the lowest dose were less likely to have used an ARB during the 6-month baseline ($p < .001$; 29% with any ARB use vs. 31-40%), which is consistent with guidelines. However, there were no significant differences in baseline ACEi use as a function of one's first dose ($p = .20$, 31%-40% with any ACEi use). Members starting on the lowest dose were not more likely to have renal disease (23% vs. 16-21%, $p = .09$) or moderate liver disease (6.7% vs. 4.4-6.5%, $p = .60$) relative to those starting on the high or medium dose, respectively.

When evaluating trajectories as a function of the dose for their final fill (Fig 1, Panel B), a large majority of those taking the low dose at their final fill remained on the low dose for their entire time on SAC/VAL. In contrast, most individuals finishing at the medium and high doses titrated up from a lower dose. Small proportions of these members required multiple titrations (up and down) before presumably stabilizing at either the medium or high dose. Members' modal dose filled typically aligned with their final dose filled (see Table 2). Because the final dose captured the predominant trajectories (stable or increase) and the most typical dosage taken, final dose filled was chosen to evaluate potential differences in dose-related outcomes (hereafter referred to as "dose group").

## Member characteristics by dose

Member characteristics are shown overall and as a function of dose group in Table 1. Members in the low dose group were older (median difference 4-6 years), more likely to be female (34% vs. 26-30%), and more likely to hold Medicare Advantage coverage (53% vs. 35-42%; $p$s ≤.001). While overall CCI scores did not reveal substantive group differences, the low dose

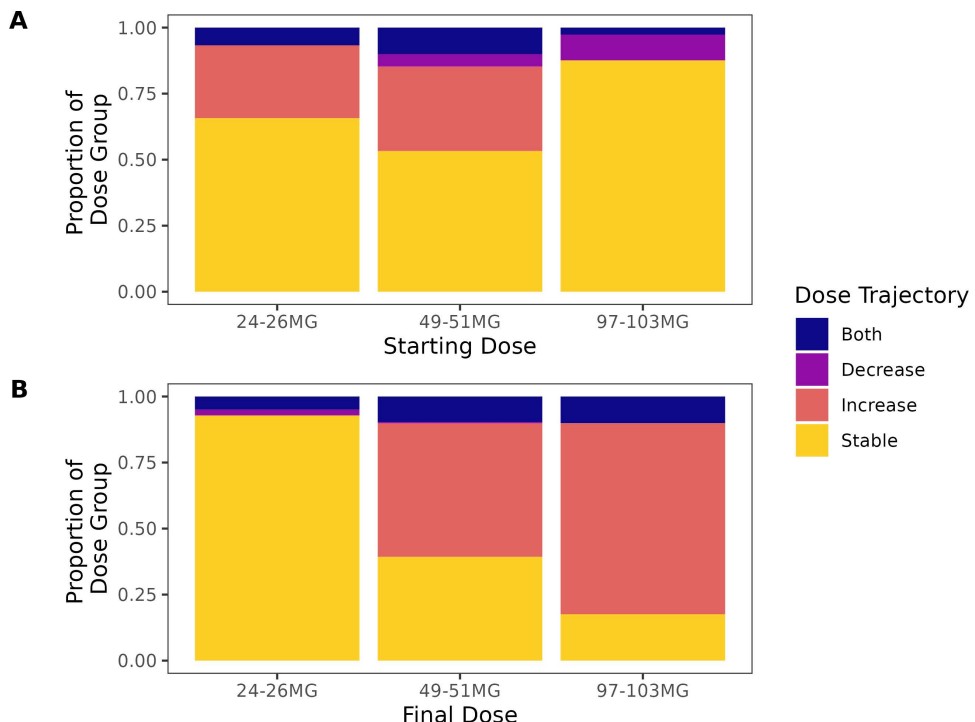

**Fig 1. Members' dose trajectories across fills by starting and final dose.** Panel A depicts the proportion of members with the various trajectories based on their first dose filled, while Panel B shows these proportions as a function of members' final dose filled.

group had a significantly higher prevalence of several individual comorbidities (e.g., peripheral vascular disease, renal disease), as well as other conditions and procedures screened for (e.g., atrial fibrillation, percutaneous coronary intervention). However, these differences were negligible to small in effect size (ranging from $V = .005$ to .10; and $\eta^2 \leq .001$ to .005). Details are provided in Table S5 in S1 File.

## Outcomes

Differences across dose groups were evident in global, but not HF-specific outcomes. Table 3 presents ratios, standard errors, and test statistics for each dose group comparison for total costs and all-cause and HF-related admissions. Full model results, including coefficient estimates for all covariates, are provided in Tables S6 to S8 in S1 File. After controlling for follow-up duration, medical, demographic, SVI, and other relevant variables, one's final SAC/VAL dose was a significant predictor of total costs of care and rates of all-cause admissions ($ps < .001$). Those ending on the highest dose incurred lower total costs of care and had fewer all-cause admissions relative to those on the low and medium doses ($ps \leq .001$), and those ending on the medium dose had significantly lower costs and fewer admissions than those on the low dose ($ps < .001$). There were no significant differences between dose groups in HF admissions ($p = .26$).

Average spending after adjusting for covariates is shown in Fig 2, both across the average duration of follow-up (Panel A) and per-member per-month (Panel B). At the month level, average costs differed across groups from $488 (medium vs. high) to $1,121 (low vs. high), which amounted to $7,089 to $16,272 across the average duration follow-up (14.5 months).

**Table 3. Model-estimated differences between dose groups presented as ratios, test statistics, and the significance thereof.**

| Outcome | Dose Group Comparison | Est. Ratio | SE | df | Test Statistic (*t/z*) | *p* |
|---|---|---|---|---|---|---|
| Total Costs | Low/Medium | 1.22 | 0.055 | 2856 | 4.50 | <.001 |
| | Low/High | 1.48 | 0.087 | 2856 | 6.59 | <.001 |
| | Medium/High | 1.21 | 0.057 | 2856 | 3.97 | 0.001 |
| All-Cause Admissions | Low/Medium | 1.52 | 0.157 | – | 4.02 | <.001 |
| | Low/High | 2.66 | 0.369 | – | 7.03 | <.001 |
| | Medium/High | 1.75 | 0.233 | – | 4.22 | <.001 |
| Heart Failure Admissions | Low/Medium | 1.24 | 0.310 | – | 0.85 | 0.674 |
| | Low/High | 1.70 | 0.542 | – | 1.65 | 0.223 |
| | Medium/High | 1.37 | 0.391 | – | 1.11 | 0.509 |

*Note.* Entries of "–" for degrees of freedom appear when a *z*-statistic is used for the given comparison. Presented ratios are relative ratios (costs) and incidence rate ratios (admissions).

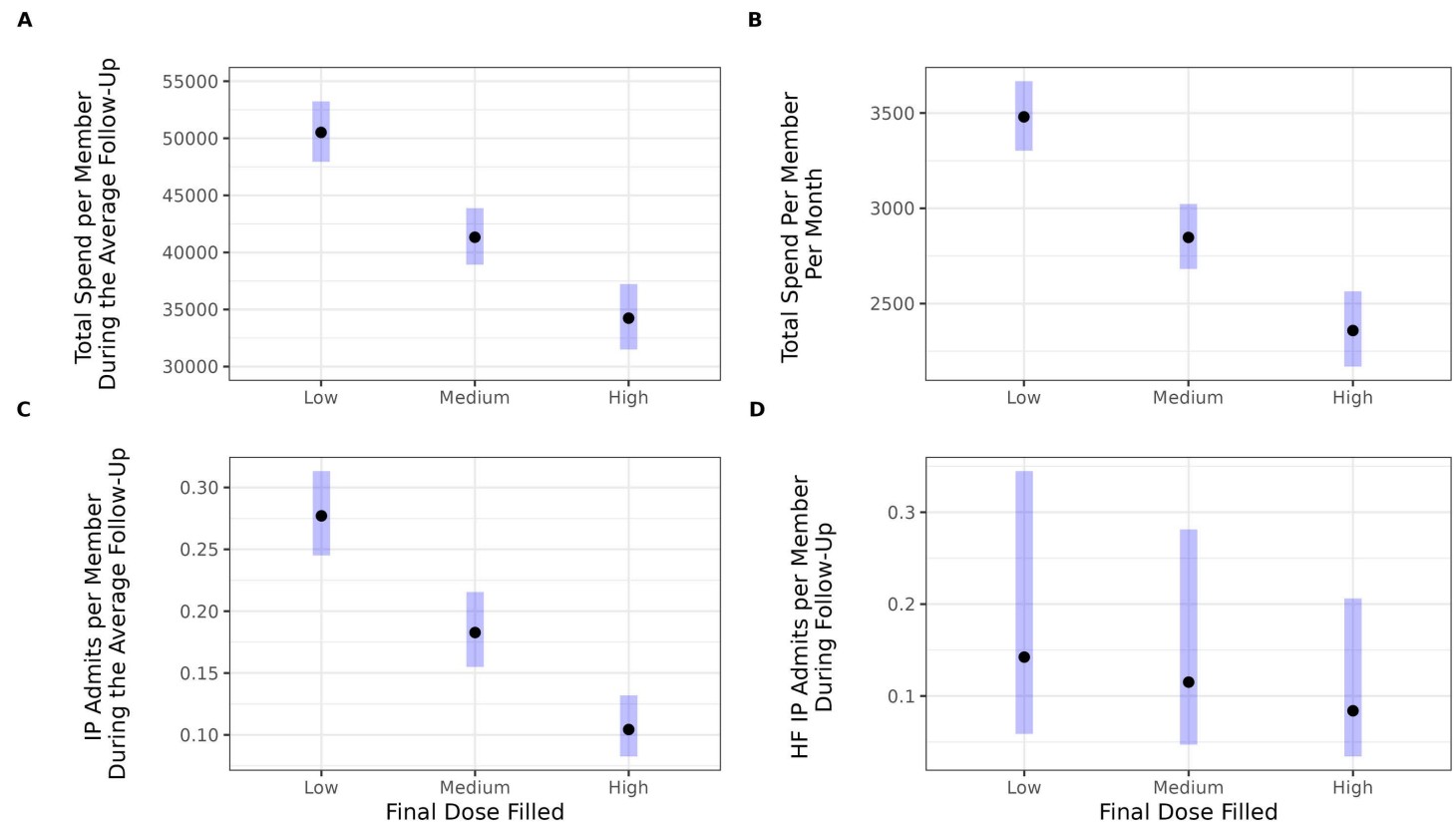

**Fig 2. Adjusted differences in outcomes (total costs, all-cause, and heart failure admissions) as a function of dose group (final dose filled) from generalized linear models.** Panels display model estimates and 95% confidence intervals for the different outcomes evaluated as a function of dose group: total costs over the average follow-up duration (14.52 months; panel A) and on a per-member per-month basis (panel B), as well as the number of all-cause (panel C) and heart failure (HF) inpatient admissions (IP; panel D) during the average follow-up (14.30 and 14.34 months for all-cause and HF, respectively).

The differences in admissions per-member during the average follow-up (14.3 months) ranged from 0.079 to 0.173 admissions. These differences were large even though admissions per-member were relatively low (see Panel C of Fig 2).

Sensitivity analyses showed a similar pattern of findings, although the differences between dose groups were larger among those taking SAC/VAL for shorter durations (3-8 months) and smaller among those taking SAC/VAL for longer durations (18 or more months) (see Table S9 in S1 File). Descriptive statistics of unadjusted cost and admission outcomes by SAC/VAL dose group and fill duration are provided overall and by time in Table S10 and Figs S2–S4 in S1 File, respectively.

## Discussion

The four major findings in this study are that most patients treated with SAC/VAL are on the lowest dose offered, dose escalation in those taking the lowest dose was uncommon, hospitalizations varied inversely with SAC/VAL dose, and the total cost of care varied inversely with SAC/VAL dosage. These findings underscore the need for a systematic re-appraisal of how this life-saving and cost-efficient medication is being used today.

PARADIGM-HF demonstrated the superiority of SAC/VAL at 200 mg bid to enalapril in the setting of a multi-center randomized controlled trial. Despite these results, real-world data suggest that only 20% of prescriptions are filled at this target dose [13]. Our findings suggest that many members with histories of ACEi and ARB use are initiated and remain on the lowest SAC/VAL dose despite limited evidence of intolerability (e.g., dose de-escalation) or contraindication (significant renal and liver function impairment at baseline). Further, an exploratory analysis comparing incidence of hypotension, hyperkalemia, and acute kidney injury across those who remained on the same dose, increased from their initial dose, or de-escalated doses during baseline and follow-up periods suggests that neither higher prevalence of these conditions in members' recent medical history, nor their occurrence following initiation appear to account for the lack titration in our sample (see coding details and additional discussion in Supplemental Methods, S1 File). Table 4 shows that prior to initiation of SAC/VAL, those who remained on the same dose and those who titrated upward had comparable, low levels of these potential contraindications; the same was true after initiation of SAC/VAL. Importantly, those who de-escalated doses during follow-up evidenced approximately 2x the rate of these complications, supporting the validity of these data for detecting potential adverse events. Together, these findings are consistent with prior work suggesting a general

**Table 4. Counts and percentages of members who had documentation of select conditions prior to initiation of SAC/VAL (Baseline) and while taking it (Follow-Up) as a function of different dose trajectories, as well as indicators of significance and effect sizes for differences across dose trajectory groups.**

| Condition | Dose Trajectory | | | $p$ | Effect Size |
|---|---|---|---|---|---|
| | Consistent dose $n = 1,907$ | Increased $n = 916$ | Down or returned to start $n = 154$ | | |
| Baseline | | | | | |
| Hypotension | 167 (8.8%) | 87 (9.5%) | 15 (9.7%) | .80 | 0.01 |
| Acute Kidney Injury | 287 (15%) | 181 (20%) | 31 (20%) | .004 | 0.06 |
| Hyperkalemia | 80 (4.2%) | 34 (3.7%) | 6 (3.9%) | .80 | 0.01 |
| Follow-Up | | | | | |
| Hypotension | 272 (14%) | 110 (12%) | 46 (30%) | <.001 | 0.11 |
| Acute Kidney Injury | 328 (17%) | 135 (15%) | 55 (36%) | <.001 | 0.12 |
| Hyperkalemia | 136 (7.1%) | 75 (8.2%) | 25 (16%) | <.001 | 0.07 |

*Note. $p$ values are from Chi-squared tests. Effect sizes are Cramer's $V$.*

underutilization of effective, evidence-based therapies for cardiac conditions in real-world settings [14–16].

The 2022 HF with reduced ejection fraction (HFrEF) therapy recommendations [3] call for the rapid initiation of four types of medications for HFrEF, including "novel" (SGLT2 inhibitors and SAC/VAL) and more traditional agents (ACEi/ARB, beta blockers and MRA). Savarese et al. recently found that rates of initiation of SGLT2 and SAC/VAL were lower than that for traditional agents at 100 days post-HF hospitalization [17]. The authors speculated this finding was due to drug side effects and "clinical inertia" [17], which is defined as a failure to initiate or intensify therapy when indicated [18]. Other studies point to the importance of cardiac specialist input. Lee et al. showed that HF patients receiving collaborative care between cardiac specialists and primary care physicians had better outcomes relative to those receiving care from primary care physicians only or none at all, which could be attributed to the greater use of guideline-recommended medications, and among other factors, hypothesized input regarding medication dosing [19]. To determine whether care from cardiac specialists may be associated with dosing patterns herein, we performed an exploratory analysis comparing cardiologist encounters across different dose titration patterns (see Supplemental Methods and Table S11 in S1 File). This analysis revealed that any SAC/VAL titration was associated with more cardiologist visits. While other interpretations cannot be ruled out (e.g., greater utilization/care-seeking is associated with more appropriate care), this result provides preliminary evidence that dose titration may be associated with provider expertise and education.

To our knowledge, our study is the first to demonstrate that treatment at higher SAC/VAL doses is associated with both fewer hospitalizations and lower total healthcare costs compared to lower doses. Additional exploratory analyses showed this same pattern of results with time to first all-cause admission as the outcome (see Supplemental Methods and Tables S12 and S13 in S1 File). Prior research evaluating both cost and health outcomes of SAC/VAL has been done irrespective of dose and relative to a propensity-matched ACEi/ARB comparison cohort [20]. Prior studies that investigated effectiveness of varying SAC/VAL doses employed much smaller samples (e.g., ≤ 721 patients per qualifying study) [7], and did not include cost data [6].

## Limitations and other considerations

This study has important strengths, including the large sample size and relatively few exclusionary conditions, but there are limitations that warrant discussion. First, our inclusion criteria and use of administrative claims yields findings that are most relevant to those who are comprehensively insured through a single payor (c.f., those who do not have prescription coverage, are enrolled in supplemental products, or have limited plan coverage). While our data source and enrollment criteria give confidence that individuals' claims will provide a broad account of their medical encounters and prescription medications, it may underestimate the magnitude of under dosing in those who are uninsured and more socioeconomically disadvantaged patient populations (i.e., those unable to maintain continuity of coverage, nor have access to certain types of coverage).

Second, given that claims data are, first and foremost, used for documentation and reimbursement of services, the accuracy of abstracted medical history and other variables dependent on diagnosis and procedure codes (e.g., heart failure admissions) may be biased or less accurate due to both variation in coding practices of billing entities, as well as how coding may be related to reimbursement and quality tracking initiatives (see [21] for an example and discussion). To this end and where possible, we used established measures and algorithms that have been tested with claims data, in addition to using encompassing coding definitions for

the variables in our data set (e.g., Charlson Comorbidity Index to characterize health status, focusing on HF diagnoses from non-diagnostic claims). We also conducted a limited chart review to gauge concordance of the SAC/VAL dosing trajectories as obtained from claims with clinical notes in members' medical records, which demonstrated perfect agreement.

Third, we used a retrospective cohort analysis, and while we statistically controlled for baseline differences, we cannot conclusively attribute spending and outcome differences to SAC/VAL dosing. Other factors, such as the quality and intensity of care could have contributed to our findings, which we could not quantify with our data source. Fourth, most members in our study resided in Western Pennsylvania. We believe it is unlikely that our results are due to region-related prescribing patterns, although we cannot confirm this.

Other outcomes, such as the overall rate of hospital admissions, are relatively low (cf. Albert et al. [20]). We believe this is due to the lack of inclusion criteria for HFrEF herein (65% had evidence of reduced EF during baseline), which is common in other reports and likely produces a more clinically severe sample. Given that the indication for SAC/VAL was recently expanded to include those with below normal (vs. low) ejection fraction, broader HF inclusion criteria are beneficial to improve our understanding of medication-associated outcomes among the more clinically diverse group that is initiating SAC/VAL. We also chose a continuous 6-month baseline period, intended to maximize the sample size, though this may have limited our ability to more accurately assess evidence of reduced EF, as well as adherence to dosing guidelines based on past ACEi/ARB use. Lastly, several of the statistical models—particularly those for costs—tended to over-predict (e.g., higher costs than actual). To counter this, we employed several approaches to ensure robustness of dose comparison findings (removal of high-influence observations, sensitivity analyses). Some specific dose differences were attenuated with longer durations of SAC/VAL use, though the patterns of differences were generally robust, and their effect magnitudes reasonable. Nevertheless, additional research is needed to understand how duration of usage may affect these dose-related differences.

## Conclusion

Our findings suggest that SAC/VAL is commonly used at doses below the recommended guidelines, which is associated with more hospitalizations and higher costs. The fact that many patients start and remain on the lowest SAC/VAL dose suggests that clinical inertia is a significant problem in patients with HFrEF and those taking SAC/VAL more generally. Guidance from the literature on clinical inertia will be essential to develop system, patient, and physician processes to address this issue.

## Supporting information

**S1 File. Word document containing supplemental methods, results, reference information pertaining to statistical software, and additional tables and figures depicting details of the sample characteristics and analysis results.**
(PDF)

## Acknowledgments

The authors thank Sarah Carey, MS, Jade Chang, and Jacalyn Newman, PhD, of Allegheny Health Network's Health System Publication Support Office (HSPSO) for their assistance in editing and formatting the manuscript. The HSPSO is funded by Highmark Health (Pittsburgh, PA, United States of America) and all work was done in accordance with Good Publication Practice (GPP3) guidelines (http://www.ismpp.org/gpp3).

## Author contributions

**Conceptualization:** Jillian M Rung, Tyson S. Barrett, Keith LeJeune, Amresh Raina, Lawrence Sinoway.

**Data curation:** Jillian M Rung.

**Formal analysis:** Jillian M Rung.

**Methodology:** Jillian M Rung, Tyson S. Barrett, Keith LeJeune, Shannon B. Richards.

**Project administration:** Keith LeJeune.

**Resources:** Shannon B. Richards.

**Software:** Jillian M Rung.

**Supervision:** Tyson S. Barrett, Keith LeJeune, Amresh Raina, Lawrence Sinoway.

**Validation:** Tyson S. Barrett, Keith LeJeune.

**Visualization:** Jillian M Rung.

**Writing – original draft:** Jillian M Rung, Tyson S. Barrett, Shannon B. Richards, Amresh Raina, Lawrence Sinoway.

**Writing – review & editing:** Jillian M Rung, Tyson S. Barrett, Keith LeJeune, Amresh Raina, Lawrence Sinoway.

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
