## [Decision Letter · Decision Letter 0]

3 Jan 2025

PONE-D-24-56892Dosing patterns and dose effects of sacubitril/valsartan: a claims-based retrospective cohort studyPLOS ONE

Dear Dr. Rung,

Thank you for submitting your manuscript to PLOS ONE. After careful consideration, we feel that it has merit but does not fully meet PLOS ONE’s publication criteria as it currently stands. Therefore, we invite you to submit a revised version of the manuscript that addresses the points raised during the review process.

We look forward to receiving your revised manuscript.

Kind regards,

Hean Teik Ong

Academic Editor

PLOS ONE

**Journal Requirements:**

**Additional Editor Comments:**

Please make minor revisions to this article in the light of the reviewers' comments:

**Reviewer 1**

This research paper reveals the real world practice of using ARNI in treating patients with heart failure. It showed the importance of up titrating of ARNI to the GDMT recommended dosage to reduce the total cost of treatment of heart failure and hospitalisation due to heart failure. The paper also analysed the likely causes of the failure of this optimisation of the therapy. The failure is most likely caused by physician’s factor (clinical inertia) as no clear explanation to the contraindications to escalate the dosage of ARNI.

Though it is a retrospective observational study with a lot of limitations as discussed it still able to send a clear message to physicians to overcome clinical inertia and adhere to clinical guidelines to help patients with heart failure to achieve better clinical outcomes and improve cost effective treatment.

**Reviewer 2**

1.Systolic heart failure is an old definition and current definition is heart failure with reduced ejection fraction. Would it be possible to classify HF by LVEF or is it not possible to obtain this data for the members?

2.The author stated that members from the database has identifiable information but the full details of the database is not accessible - Please explain further the implications of this statement, and why this database is nevertheless still valid academically and statistically for this review.

Reviewers' comments:

Reviewer's Responses to Questions

**Comments to the Author**

1. Is the manuscript technically sound, and do the data support the conclusions?

Reviewer #1: Yes

Reviewer #2: Yes

2. Has the statistical analysis been performed appropriately and rigorously? 

Reviewer #1: Yes

Reviewer #2: I Don't Know

3. Have the authors made all data underlying the findings in their manuscript fully available?

Reviewer #1: Yes

Reviewer #2: No

4. Is the manuscript presented in an intelligible fashion and written in standard English?

Reviewer #1: Yes

Reviewer #2: Yes

5. Review Comments to the Author

**Reviewer #1:**  This research paper reveals the real world practice of using ARNI in treating patients with heart failure. It showed the importance of up titrating of ARNI to the GDMT recommended dosage to reduce the total cost of treatment of heart failure and hospitalisation due to heart failure. The paper also analysed the likely causes of the failure of this optimisation of the therapy. The failure is most likely caused by physician’s factor (clinical inertia) as no clear explanation to the contraindications to escalate the dosage of ARNI. Though it is a retrospective observational study with a lot of limitations as discussed it still able to send a clear message to physicians to overcome clinical inertia and adhere to clinical guidelines to help patients with heart failure to achieve better clinical outcomes and improve cost effective treatment.

**Reviewer #2:**  Systolic heart failure is an old definition and current definition is heart failure with reduced ejection fraction. Would it be possible to classify HF by LVEF or is it not possible to obtain this data for the members?

6. PLOS authors have the option to publish the peer review history of their article (what does this mean? ). If published, this will include your full peer review and any attached files.

**Do you want your identity to be public for this peer review?** For information about this choice, including consent withdrawal, please see our Privacy Policy .

Reviewer #1: No

Reviewer #2: No

---

## [Author Response · Author response to Decision Letter 1]

3 Feb 2025

We have responded to the reviewers' comments in the attached response to reviewers document. We extend our sincere appreciation for their time and helpful feedback, which has resulted in a more complete consideration of study limitations, and provided opportunity to highlight important considerations with claims data.

---

## [Decision Letter · Decision Letter 1]

16 Feb 2025

Dosing patterns and dose effects of sacubitril/valsartan: a claims-based retrospective cohort study

PONE-D-24-56892R1

Dear Dr. Rung,

We’re pleased to inform you that your manuscript has been judged scientifically suitable for publication and will be formally accepted for publication once it meets all outstanding technical requirements.

Kind regards,

Eyob Alemayehu Gebreyohannes, PhD

Academic Editor

PLOS ONE

Additional Editor Comments (optional):

Reviewers' comments:

Reviewer's Responses to Questions

**Comments to the Author**

1. If the authors have adequately addressed your comments raised in a previous round of review and you feel that this manuscript is now acceptable for publication, you may indicate that here to bypass the “Comments to the Author” section, enter your conflict of interest statement in the “Confidential to Editor” section, and submit your "Accept" recommendation.

Reviewer #1: All comments have been addressed

Reviewer #2: All comments have been addressed

2. Is the manuscript technically sound, and do the data support the conclusions?

Reviewer #1: Yes

Reviewer #2: Yes

3. Has the statistical analysis been performed appropriately and rigorously? 

Reviewer #1: Yes

Reviewer #2: Yes

4. Have the authors made all data underlying the findings in their manuscript fully available?

Reviewer #1: Yes

Reviewer #2: Yes

5. Is the manuscript presented in an intelligible fashion and written in standard English?

Reviewer #1: Yes

Reviewer #2: Yes

6. Review Comments to the Author

Reviewer #1: This paper gives supporting evidence that optimising GDMT for heart failure patients is an important component to achieve good clinical outcomes.

Reviewer #2: As it was not possible to get information on ejection fraction, the definition of reduced ejection fraction heart failure has been inferred from diagnostic codes associated with mildly reduced or reduced ejection fraction. This is the group of HF patients with clear benefit with sacubitril valsartan.

7. PLOS authors have the option to publish the peer review history of their article (what does this mean? ). If published, this will include your full peer review and any attached files.

**Do you want your identity to be public for this peer review?** For information about this choice, including consent withdrawal, please see our Privacy Policy .

Reviewer #1: No

Reviewer #2: No

---

## [Editor Report · Acceptance letter]

PONE-D-24-56892R1

PLOS ONE

Dear Dr. Rung,

I'm pleased to inform you that your manuscript has been deemed suitable for publication in PLOS ONE. Congratulations! Your manuscript is now being handed over to our production team.

Kind regards,

on behalf of

Dr. Eyob Alemayehu Gebreyohannes

Academic Editor

PLOS ONE
